# Harnessing the Immunological Effects of Radiation to Improve Immunotherapies in Cancer

**DOI:** 10.3390/ijms24087359

**Published:** 2023-04-16

**Authors:** Gary Hannon, Maggie L. Lesch, Scott A. Gerber

**Affiliations:** 1Department of Surgery, University of Rochester Medical Center, Rochester, NY 14642, USA; gary_hannon@urmc.rochester.edu (G.H.); maggie_lesch@urmc.rochester.edu (M.L.L.); 2Center for Tumor Immunology Research, University of Rochester Medical Center, Rochester, NY 14642, USA; 3Wilmot Cancer Institute, University of Rochester Medical Center, Rochester, NY 14642, USA; 4Department of Microbiology and Immunology, University of Rochester Medical Center, Rochester, NY 14642, USA

**Keywords:** radiation, cancer, immunity, immunotherapies, immunogenic cell death

## Abstract

Ionizing radiation (IR) is used to treat 50% of cancers. While the cytotoxic effects related to DNA damage with IR have been known since the early 20th century, the role of the immune system in the treatment response is still yet to be fully determined. IR can induce immunogenic cell death (ICD), which activates innate and adaptive immunity against the cancer. It has also been widely reported that an intact immune system is essential to IR efficacy. However, this response is typically transient, and wound healing processes also become upregulated, dampening early immunological efforts to overcome the disease. This immune suppression involves many complex cellular and molecular mechanisms that ultimately result in the generation of radioresistance in many cases. Understanding the mechanisms behind these responses is challenging as the effects are extensive and often occur simultaneously within the tumor. Here, we describe the effects of IR on the immune landscape of tumors. ICD, along with myeloid and lymphoid responses to IR, are discussed, with the hope of shedding light on the complex immune stimulatory and immunosuppressive responses involved with this cornerstone cancer treatment. Leveraging these immunological effects can provide a platform for improving immunotherapy efficacy in the future.

## 1. Introduction

Ionizing radiation (IR) is a prevalent non-surgical intervention for achieving cures in cancer [1]. Approximately 50% of cancers are treated with some form of IR [2,3], and it is estimated that this contributes to 40% of cures [4,5]. While the cytotoxic effects of IR on cancer were first appreciated in the early 20th century, following Röntgen’s discovery of X-rays in 1895 [6], the complex mechanisms by which the tumor microenvironment (TME) is altered following IR are still being elucidated today [7,8,9].

Notably, the immune modulatory potential of IR has gained particular interest. Cancers exploit naturally occurring immunosuppressive mechanisms to facilitate their growth and metastasis, and many reports show that IR is capable of reprogramming the TME to alleviate this suppression and even achieve abscopal effects in rare cases [10]. Following IR exposure, cancer cells undergo an immunogenic form of cell death which results in antigen presenting cell (APC) activation and maturation, leading to eventual cytotoxic T cell responses specific to the tumor [11]. Interestingly, multiple reports now describe that an intact immune system is essential to the efficacy of IR [12,13,14,15].

While the large body of evidence for the immune stimulatory effects of IR is certainly convincing, numerous reports also show that IR can induce a wound healing response characterized by tumor-associated macrophage (TAM), myeloid-derived suppressor cell (MDSC) and regulatory T cell (Treg) recruitment to tumors [16,17]. These cells dampen anti-tumor immune responses and initiate angiogenic processes, augmenting tumor growth and expansion [18,19]. Moreover, a higher number of these cells correlate with poorer prognosis in many cancers [20,21]. Hence, a dichotomy exists with this therapy where both immune stimulatory and immunosuppressive effects occur, often simultaneously [16].

Navigating this literature can be challenging, as the immunological responses are highly dynamic and can vary depending on the TME in question and the IR protocol used [22,23,24,25,26]. Here, we provide an overview on the effects of IR on the immune components of the TME. The influence of IR on immunogenic cell death (ICD) will be discussed, along with the dynamic responses associated with both myeloid and lymphoid immune subsets, providing a greater overall picture of the complex immune stimulative and suppressive processes elicited by IR in tumors. Harnessing the immunological effects of IR is critical to optimizing the efficacy of combination immunotherapies in the future.

## 2. Immunogenic Cell Death

Immediately following radiation, DNA is damaged via direct ionization or indirectly through the action of free radicals (such as reactive oxygen species; ROS) generated when radiation ionizes water molecules or biomolecules in cells. Cells are composed of 80% water, and it has been estimated that the indirect effects of radiation account for 60% of the total cellular damage accumulated [27]. This DNA damage occurs in the form of single-strand and double-strand breaks that are sensed by ataxia telangiectasia mutated (ATM) and ATM- and RAD3-related (ATR) kinases, which in turn activate DNA repair mechanisms in the cells. However, if the damage accumulated exceeds cellular repair capabilities, the cells will undergo cell death or senescence [28].

### 2.1. Senescence and Cell Death

In the case of senescence, the cell terminates division processes through p53/p21 and p16/RB1 signaling [29]. The cell remains viable but undergoes arrest to prevent further damage and outgrowth of mutated cells. For cell death, on the other hand, several death pathways can occur depending on the extent of the damage. Lower, recoverable levels of damage are associated with programmed mechanisms of death such as apoptosis and autophagy, whereas higher levels evoke irreparable damage resulting in necrosis [30]. Typically, necrosis has been defined as immunogenic and apoptosis has been defined as immune silent; however, it is now known that all forms of cell death possess some level of immunogenicity [31]. ICD broadly refers to a form of cell death that involves the release of damage-associated molecular patterns (DAMPs), chemokines, cytokines and tumor antigens that prime APCs and instigate adaptive immune responses (Figure 1) [11]. IR has been shown to induce these markers of ICD (calreticulin, high mobility group box 1 (HMGB1) and adenosine triphosphate (ATP)) in a dose-dependent manner (ranging from 0 to 100 Gy) [32,33].

### 2.2. Damage-Associated Molecular Patterns

Following IR-induced cellular damage, DAMPs such as calreticulin, heat shock protein-70 (HSP70), HMGB1 and ATP, which are normally endogenous in nature, get released by dying cells to facilitate phagocytosis [31,34]. Calreticulin and HSP70 are endoplasmic reticulum (ER) molecular chaperones that translocate to the cell membrane following cellular damage. Once expressed on the surface, they can bind to CD91 expressed on APCs and act as an ‘eat me’ signal for the dying cell [35,36]. Additionally, the HSP family of proteins may also present carried tumor antigens to APCs directly [37]. HMGB1 is a nuclear protein that governs chromosomal structure and function. Although its main function is to act as a DNA chaperone, it can be passively released during cell death and bind to toll-like receptors (mainly TLR4) on APCs to augment phagocytosis and antigen processing via MyD88 signaling [38,39]. ATP is somewhat different, as it requires autophagy for its active release from dying cells [40]. Autophagy is a stress response that initiates the molecular recycling of cellular organelles and cytoplasmic proteins under hostile conditions. Following extracellular release, ATP functions as a chemoattractant for dendritic cells (DC) by binding to purinergic receptors [41,42]. Conversely, although DAMPs generation in the tumor is an important immune activator, studies assessing correlations between the levels and overall outcome have generated contested results [43,44]. It is likely that elevating DAMPs alone is not enough to elicit measurable benefits to radiation, but many factors of the tumor microenvironment (TME) are critical when considering this response.

### 2.3. Cytokine Release

The release of cytokines following IR is a massive field of research, with a large volume of work being published on various cytokines including IFN-γ [45], IL-1β [46], TNF-α [47], IL-6 [48] and TGF-β [49]. Among these, the Type-I interferons (IFN) have proven particularly prominent in recent times. Type-I IFN is a family of pro-inflammatory cytokines heavily involved in the response to viral infection. They are also pivotal for anti-tumor immune responses following IR, primarily through DC cross-priming [50,51,52]. IR induces Type-I IFN expression (IFN-α and IFN-β, in particular) through multiple endogenous nucleic acid sensing pathways following the release of double-stranded DNA and RNA into the cytosol of damaged cells [53]. Cyclic GMP-AMP synthase (cGAS) is an enzyme that catalyzes the formation of cyclic GMP-AMP (cGAMP) when it recognizes cytosolic DNA. cGAMP then binds to and activates STING (stimulator of interferon genes) in the ER [54,55]. STING is an adaptor molecule, and its activation leads to downstream Type-I IFN expression [56]. Notably, STING also activates NF-κB signaling, inducing a vast array of proinflammatory cytokines such as TNF-α, IL-1β and IL-6 [57,58]. DNA and RNA are also capable of triggering Type-I IFN release through endosomal TLRs (TLR9 for DNA; TLR3, 7 and 8 for RNA) and their adaptor molecules MyD88 or TRIF [59,60]. Cytosolic RNA can additionally be detected by RIG-I-like receptors (RLR) and induce IFN-β expression via the mitochondrial adaptor protein MAVS [61]. Regardless of the pathway involved, however, multiple reports have demonstrated that Type-I IFN production is essential to IR efficacy. When this pathway is inhibited, tumor burden is similar to unirradiated controls, and it has been shown that CD8^+^ T cell cytotoxic function in these tumors is impaired [52,62,63,64].

### 2.4. Chemokine Release

IR leads to an influx of immune cells into the TME through chemokine gradients as a form of damage control. Chemokines are produced by multiple cells in the TME and bind to their cognate receptors in an autocrine or paracrine fashion [65]. CCL2 is upregulated in tumors following IR, and this facilitates CCR2^+^ monocyte and Treg recruitment to tumors [66,67]. Recent research points towards STING-dependent expression of CCL2, suggesting this pathway is a double-edged sword in IR treatment [68]. On one hand, there is Type-I IFN expression that fosters anti-tumor immunity; on the other, there is an influx of immunosuppressive monocytes and Tregs that evoke radioresistance and tumor growth [69]. Another chemokine ligand produced downstream of the cGAS-STING pathway following IR is CCL5 [70]. CCL5 is also involved in trafficking monocytes to the tumor utilizing the CCR5 receptor, in particular [70,71]. While the receptors for these chemokines are also expressed on T cells, higher levels of CCL5 expression are typically associated with poorer outcomes in cancer [72]. This suggests that the immunosuppressive infiltrate dominates any potential anti-tumor immune responses driven by chemokine signaling following IR. Overall, the broad topic of chemokine expression following radiation is highly complex, influencing both pro- and anti-tumor immune cell types and varying based on the tumor, IR dose and time-point measured [73,74,75]. It has been shown, however, that inhibiting CCR2/CCR5 signaling can sensitize tumors to IR [71].

**Figure 1 ijms-24-07359-f001:**
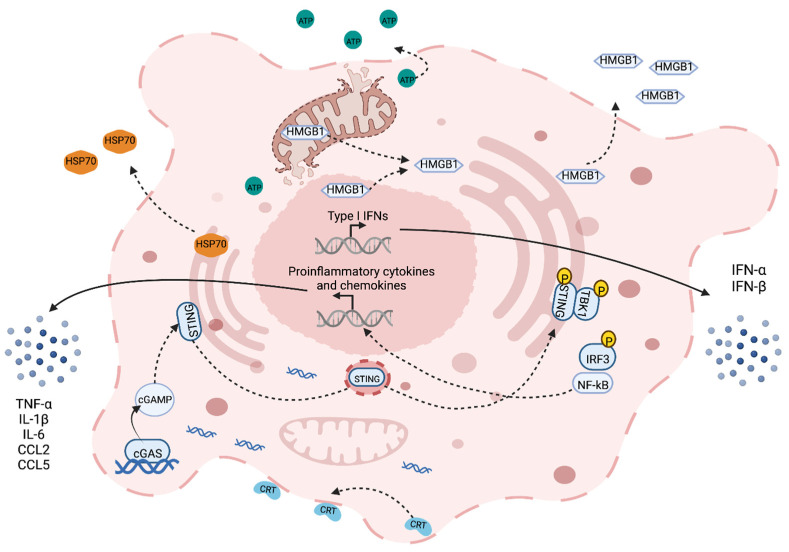
Immunogenic cell death elicited by IR. ICD induced by IR is marked by the release of DAMPs (HMGB1, double-stranded DNA and RNA, HSP70, ATP and calreticulin (CRT)), Type I IFN (IFN-α and IFN-β) through cGAS-STING signaling and other cytokines (TNF-α, IL-1ß and IL-6) and chemokines (CCL2 and CCL5) through STING-induced NF-κB activation. Broken lines indicate movement and continuous lines represent the production of the given target.

## 3. Myeloid Responses to Radiation

As a consequence of IR-induced ICD, the released DAMPS, tumor antigens, cytokines and chemokines recruit and differentiate various myeloid subsets in the TME. Myeloid cells originate from hematopoietic stem cells in the bone marrow and circulate in the blood and lymphatic system, ready to respond to tissue damage or infection [76]. They represent a branch of leukocytes and consist of monocytes, macrophages, DCs and granulocytes [77]. These cells may already exist in high numbers in the tumor milieu, but they can also be re-directed from the blood to the tumor site in response to IR [78]. Depending on the tumor’s propensity for immune evasion, the myeloid cells may exhibit a pro- or anti-tumor phenotype based on the cytokine and chemokine profile of the tumor, along with the expression of immunosuppressive factors [79].

### 3.1. DCs and Other APCs

DCs are rare immune cells within tumors but the most potent APCs for initiating adaptive immune responses. These cells sample the TME for antigens that get processed endogenously and presented to CD8^+^ and CD4^+^ T cells in the tumor draining lymph node to mount tumor-specific immune responses [80].

Tumor-residing and tumor-infiltrating DCs respond strongly to the release of DAMPs, tumor antigens and Type-I IFN released into the TME following IR. As mentioned above, ICD results in the release of many DAMPs that can bind to DCs directly though surface receptors or present the tumor antigen through their chaperon functionality. ATP can bind directly to the purinergic P2RX_7_ receptors on DCs and elicit inflammasome activation, leading to pro-inflammatory cytokine induction [81]. Calreticulin and HSP70 bind to the receptor CD91—among others—on DCs and elicit phagocytosis [82,83]. Triggering CD91 on APCs with these DAMPs has also been shown to activate NF-κB signaling and induce pro-inflammatory cytokine and chemokine expression, leading to various CD4^+^ T cell responses [84]. Both double-stranded nucleic acids and HMGB1 released from damaged cells trigger TLR signaling on APCs. Regardless of the TLR triggered, Type I IFN signaling is activated, which induces APC maturation, migration to the tumor- draining lymph node and antigen presentation [53,85]. Importantly, this Type-I response is also critical for monocyte differentiation to DCs in the TME and the cross-presentation of the antigen for CD8^+^ T cell activation [86,87,88]. IFN-α can bind to IFNα/β receptors on monocytes and induce the expression of DC markers (CD80, CD86, CD40, MHC II), initiating a pro-inflammatory phenotype in these cells [89,90,91]. Additionally, IFN-α could also prolong the life of internalized antigens in DCs, encouraging greater levels of cross-presentation [92]. Therefore, IR plays an influential role with regard to DC activity in the TME: from DC infiltration and differentiation to eliciting antigen uptake, cross-presentation and fostering pro-inflammatory responses.

### 3.2. MDSCs and TAMs

MDSCs are a diverse population of immature myeloid cells that possess strong immunosuppressive functions. These cells generally accumulate at sites of chronic inflammation, where they mitigate sources of inflammation and promote angiogenesis as a wound healing response [93]. This process can also be hijacked by tumors, however, to dampen anti-tumor immune responses and promote cancer growth and metastasis [94]. TAMs also serve as a source of immune evasion for tumors. Macrophages are large phagocytes that internalize cellular debris and contribute to wound-healing responses. Macrophages can be immunosuppressive or stimulatory depending on the context of their local environment, and in cancer, this is typically an immunosuppressive phenotype [95]. Overall, higher levels of MDSCs and TAMs are associated with a poor overall outcome across multiple cancer types [96,97,98]. Notably, higher doses of IR (≥10 Gy/fraction or cumulative doses up to 60 Gy) have been most associated with this chronic inflammatory/immunosuppressive response [25,99].

As IR subjects tumors to tissue damage, it is no surprise that this coincides with MDSC and TAM accumulation in the tumor [68,100]. IR induces upregulations of chemokines that facilitate the recruitment of these cells to the TME, where they are differentiated to a ‘pro-tumor’ phenotype and exert an array of immunosuppressive functions [101]. Notably, these cells are potent producers of the immunosuppressive cytokines IL-10 and TGF-β. IL-10 inhibits APC activity by downregulating the expression of MHC II [102,103]. It also has profound antagonistic effects on pro-inflammatory cytokine and chemokine signaling in the TME, encouraging an anti-inflammatory milieu associated with aggressive tumors [96,104]. TGF-β plays a similar role in tumors, polarizing CD4^+^ T cells to immunosuppressive Tregs and macrophages to an anti-inflammatory phenotype [105,106]. These pro-tumor TAMs can secrete the enzyme arginase that depletes the amino acid L-arginine in the TME, a key metabolite involved in T cell regulation [107,108]. In addition, MDSCs are sources of reactive oxygen species (ROS) which also impair T cell responses [109,110]. Hence, the influx of TAMs and MDSCs to the TME in response to IR-induced tissue damage can counteract the initial innate and adaptive immune responses stimulated through Type-I IFN signaling. This has also been shown to be a key mechanism involved in the development of radioresistance [68]. Efforts to re-wire these cells to a more ‘anti-tumor’ phenotype have shown promise in sensitizing cancer to IR [111].

### 3.3. Neutrophils

Neutrophils are the most abundant granulocyte and are typically the first line of defense against infection and inflammation. Like TAMs, tumor-associated neutrophils (TANs) can be ‘pro’ or ‘anti-tumor’ depending on the context of the TME. In most cases, however, higher numbers are associated with a poor overall outcome [112,113]. They are also key players in acquired radioresistance, associated with their prevalent wound-healing phenotype [114,115]. Neutrophils are among the first cells recruited to the tumor following IR. In their anti-tumor phenotype, they produce ROS and elicit cancer cell apoptosis [116,117]. By contrast, they can also promote tumor growth and angiogenesis via their wound-healing features. They are also a source of arginase, which, along with ROS, antagonizes T cell activity [118]. CXCL1, 2 and 5 are readily produced by tumors and facilitate neutrophil recruitment to tumors via their CXCR1/2 receptor [119]. Once there, immunosuppressive cytokines such as TGF-β polarize TANs to a pro-tumor phenotype [120]. IR has been shown to aggravate this pro-tumor phenotype. Notably, IR induced the formation of neutrophil extracellular traps (NETs; web-like structures consisting of DNA and histones) that promoted treatment resistance in preclinical models of bladder cancer [121]. Overall, IR drives neutrophil recruitment to tumors that get polarized in the TME to a ‘pro-tumor’ phenotype and promote radioresistance through wound-healing processes.

## 4. Lymphoid Responses to Radiation

Lymphoid cells also originate from hematopoietic stem cells in the bone marrow and develop into T cells, B cells or NK (natural killer) cells. This maturation occurs in the thymus for T cells, in the bone marrow and spleen for B cells and in the secondary lymphoid tissues for NK cells [122,123,124]. There is also evidence to suggest that a subset of DCs may derive from this lineage as well [125]. These cells represent branches of the innate and adaptive immune system and are responsible for the cell-killing of cancer or infected cells, regulating immune responses and antibody production [117,126].

### 4.1. T Cells

The release of tumor antigens and APC activation after IR are key events in the activation of CD8^+^ and CD4^+^ T cells. It has been demonstrated that T cells are required for IR efficacy [12,127,128]. CD8^+^ T cells are the most prominent anti-tumor immune cells capable of directly killing cancer cells via the recognition of peptides on MHC I molecules expressed on cancer cells and APCs [129]. CD4^+^ T cells are a heterogeneous subset of immune stimulatory or immunosuppressive lymphocytes discriminated by their cytokine secretion profiles. In their anti-tumor state, these cells can bind to peptides presented on MHC II molecules expressed on APCs and promote CD8^+^ T cell cytotoxic function through the secretion of activating cytokines such as IL-2 and IL-21 [130]. Recent research has shown that stereotactic body radiation therapy (SBRT)—a form of ablative IR that consists of higher doses over shorter fractions than conventional IR—can elicit the clonal expansion of T cell receptors in cancer [131,132,133]. This indicates that the tumor antigens released by IR result in a more precise T cell anti-tumor immune response, as the T cells with receptors specific to tumor antigens experience an outgrowth. Indeed, this has provided a strong rationale for combining SBRT (typically ≥5 Gy oligofractions [25,132,134,135]) with immunotherapies to overcome immunosuppressive mechanisms in the TME and take full advantage of T cell-specific responses in multiple cancers [136]. On top of this increased antigen release, IR has also been shown to increase MHC I expression on cancer cells, providing a greater pool of antigens for CD8^+^ T cells to sample [137]. Type-I IFN signaling has been suggested as the regulator of IR-induced MHC I [138], and a recent study pointed towards nod-like receptor C5 (NLRC5) signaling, a transactivator of MHC I [139]. Similarly, the proinflammatory cytokine IFN-γ has also shown potential to upregulate MHC I following IR [45]. IFN-γ is also responsible for CD8^+^ T cell trafficking to the TME following IR and maintaining their cytotoxic capacity [13,45]. Moreover, inhibiting this cytokine abrogated the IR efficacy of preclinical models of melanoma, colorectal and pancreatic cancer [13,45,111].

By contrast, IR has also been shown to suppress T cell anti-tumor immune responses by various means. Notably, immune checkpoint proteins (PD-L1, in particular) become upregulated in response to IR via Type I and Type II IFN signaling [140,141,142]. Immune checkpoints are naturally occurring ‘breaks’ in immune responses that prevent the overstimulation of immune responses and damage to the host [143]. These breaks are hijacked in cancer, however, as a mechanism of overcoming the host’s anti-tumor immune response [144]. PD-L1 expressed on cancer cells can bind to its receptor (PD-1) on T cells and terminate T cell receptor signaling, preventing adaptive immune responses [144]. Many clinical trials are ongoing, combining IR with immune checkpoint inhibitors to counteract this problem [145,146]. On the same note, CD4^+^ T cells in the TME can be polarized to immunosuppressive Tregs by MDSCs and TAMs and exert similar functions on adaptive immunity [147]. Tregs also act as breaks on the immune system through immune checkpoint expression (CTLA-4 and PD-1), along with immunosuppressive cytokine (IL-10 and TGF-β) and metabolite (adenosine) secretion that affect a wide range of anti-tumor immune responses [148]. Trials are ongoing to target multiple pathways involved in Treg modulation to anti-tumor immunity and improve IR efficacy [149,150].

### 4.2. NK Cells

NK cells are innate lymphocytes that do not need to encounter an antigen to mount cytotoxic responses. They lack expression of a T cell receptor but have alternative activating receptors (e.g., NKG2D and NKp80) that, once ligated, can result in the direct killing of the cell or pathogen [151]. This is particularly useful in cancer, as they do not require multiple signals prior to eliciting their cytotoxic and immune stimulatory effects [152]. A recent study in pancreatic cancer patients showed that IR induced elevated tumor levels of NK cells through the chemokine CXCL8 that binds to CXCR1 and CXCR2 in human NK cells. This upregulation was found to be dependent on NF-κB signaling and was positively correlated with IR and cetuximab treatment responses [153]. Preclinical models of oral squamous cell carcinoma also showed that NK cell prevalence was increased in tumors following IR and were associated with a more activated phenotype, upregulating activation receptors and cytokine expression. It was found that these cells were critical to the efficacy of IR and ATR inhibition in this model [154]. Multiple clinical studies have shown that the circulatory numbers and activation of NK cells increase for months post-IR [155,156]. Despite this, there are reports describing no change to intratumoral NK cell density and even reduced sensitivity to NK cell-mediated cytotoxicity following IR [111,157,158]. NK cells have recently been shown to express the immune checkpoint PD-1, making it potentially sensitive to IR-induced overexpression of PD-L1 on cancer cells [159]. Curiously, while the loss of MHC I expression is a positive cytotoxic signal for NK cells, the lack of MHC I expression in tumors has not been shown to correlate with enhanced NK cytotoxicity. These findings suggest that alternative immunosuppressive mechanisms may abrogate NK activity in tumors [160,161,162]. Hence, the literature is conflicting regarding NK cell activity in tumors following IR. While some reports describe increased infiltration and activity after IR, which may predict treatment outcomes, the immune landscape of the TME plays a considerable role in NK cell functionality.

### 4.3. B Cells

B cells are central players to the humoral immune system. After maturing in the bone marrow, these cells circulate through secondary lymphoid tissues in search of cognate antigen and secondary signals that evoke phenotypic changes in the cell producing memory B or plasma cells. These cells are responsible for antibody production and regulating T cell and NK cell function [163,164]. This regulation occurs through antigen presentation and antibody and cytokine production [165,166]. In mouse models, up to one-third of the cells in tumor-draining lymph nodes are B cells, emphasizing their role in adaptive immunity [167]. The levels of B cell and plasma cell gene signatures have been shown to correlate with better overall survival in numerous cancers [166]. By contrast, B cells are also known to be heavily immunosuppressive [168,169]. Regulatory B cells (Bregs) and IL-10-producing B cells (B10s) antagonize anti-tumor immunity through similar Treg-related mechanisms: IL-10, TGF-β and PD-L1 expression [170,171]. IR (SBRT, in particular) has been shown to increase B cell numbers up to 14 days post-treatment in subcutaneous colon tumors. The same was also observed for T cells and macrophages in this model [172]. Interestingly, depleting B cells in preclinical models of head and neck squamous cell carcinoma resulted in a loss of local control following treatment with IR and anti-PD-L1, suggesting a positive role in treatment response in this case. This same study demonstrated that IR induced B cell maturation and increases in tumor antigen-specific B cells [173]. Besides this, little work has been published on the effects of IR on B cells. We know that B cells facilitate T cell and NK cell function in tumors and that their numbers may increase in tumors following IR. However, future work is required to decipher changes to B cell receptor sequencing and cytokine and immune checkpoint expression following IR.

## 5. Enhancing Anti-Tumor Immune Responses with IR

In rare cases, IR has demonstrated potential for systemic immune responses, attenuating the growth of distal tumors not directly subjected to the treatment. A meta-analysis published in 2016 counted a total of 46 abscopal effects reported in the literature between 1969 and 2014 [174]. The adaptive immune system is likely responsible for this infrequent phenomenon, as immune memory generated by antigens presented from the treated site can also be expressed elsewhere [22]. This is in line with the finding that the most immunogenic cancers harbor the greatest mutational burden; in other words, more mutations correspond to a greater number of tumor antigens for the adaptive immune response to take advantage of [175,176,177]. This has led to a huge surge in interest in the Type-I IFN response induced by IR, as the ability of this pathway to drive antigen presentation has found new significance in light of targeting metastatic disease [10,178]. Two exciting treatment avenues have emerged as a result of this effect: (1) combining immunotherapies with IR and (2) targeting both the primary and metastatic tumors with IR.

### 5.1. IR and Immunotherapy

Combining IR with immunotherapies in an attempt to achieve abscopal effects has gained huge momentum in the last decade. Throughout this text, we have described immunosuppressive mechanisms that are currently being targeted in the clinic to enhance the anti-tumor immune response generated by IR. Clinical trials with this combination therapy are reviewed extensively elsewhere but include immune checkpoint inhibitors (PD-1, PD-L1, TIGIT and CTLA-4), cytokines (IL-2), DC maturation agonists (GM-CSF) and T cell activation agonists (anti-CD27 and anti-CD40) [179]. Notably, the IR doses used vary greatly across these trials (2–8 Gy fractions of 24–66 Gy total dose). Multiple reports of clinical abscopal effects are now being described across an array of cancers with this regime, providing a potential treatment option for advanced cancers that, until now, had no options but palliative care [178,180,181].

### 5.2. RadScopal Effect

Expanding on this treatment, it has been suggested that IR delivered directly to a metastatic lesion may enhance the abscopal effects generated with immunotherapies and IR to the primary tumor [182,183]. Targeting metastatic disease may release antigens not associated with primary tumors, enhancing effector responses systemically. This has been referred to as the ‘RadScopal effect’ and is currently being trialed in multiple cancers [184]. With this treatment, the primary tumor is treated with the conventional IR regime alongside immune checkpoint inhibitors. In an effort to boost this response, secondary lesions are also targeted with low-dose fractions (1–2 Gy) [185]. These low-dose interventions are thought to increase T cell and DC infiltration to the lesions without inducing a complementary myeloid influx associated with the above-mentioned wound-healing response [186]. The regime has been proven capable of curing metastatic models in vivo, and several clinical studies have noted strong growth reductions to both treated and untreated metastatic sites [182,185,186,187]. This innovative multi-site IR treatment provides tentative hope for treating metastatic disease in the future.

## 6. Conclusions

The effects of IR on the immune landscape of tumors are comprehensive and dynamic (Figure 2 and Table 1). ICD induced by IR provides a platform for myeloid and lymphoid cell subsets to infiltrate the tumor and evoke a range of pro-tumor and anti-tumor effects. Leveraging these responses to force an anti-tumor immune response is key to improving treatment responses in the future and reaching IR’s full therapeutic potential. To this end, combination immunotherapies and the innovative targeting of secondary legions with IR have broadened the potential disease stage that can be targeted effectively, providing hope for an advanced disease that has no current treatment options.

## Figures and Tables

**Figure 2 ijms-24-07359-f002:**
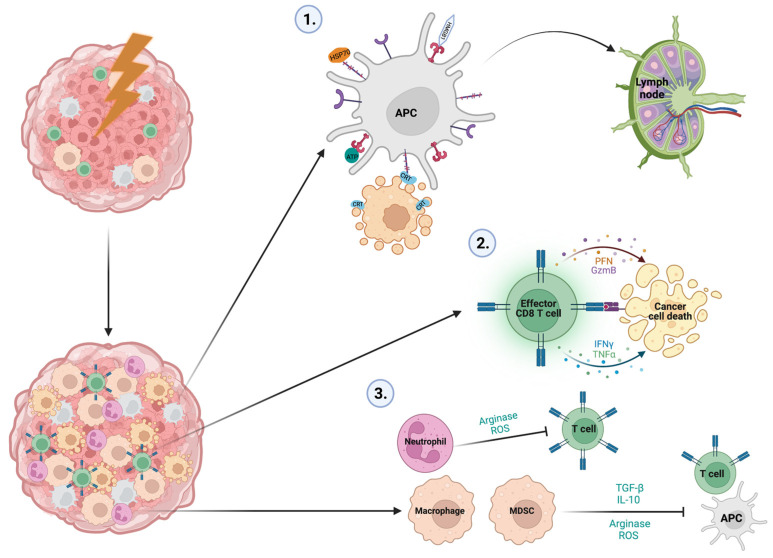
Key immune stimulatory and immunosuppressive events associated with IR treatment in cancer. Following IR, ICD results in the release of DAMPs, cytokines, chemokines and tumor antigens which activate APCs and induce the mass infiltration of myeloid and lymphoid cells. Three key immunological events occur following ICD which dictate the treatment response: (1) APCs internalize, process and present tumor antigens released by damaged tumor cells and migrate to the tumor draining lymph node to present these antigens to CD4^+^ and CD8^+^ T cells. (2) CD8^+^ T cells with receptors specific to the tumor antigen localize to the tumor and kill cancers via the cytolytic enzymes perforin and granzyme B, along with the cytotoxic cytokines IFN-γ and TNF-α. (3) Immunosuppressive macrophages and MDSCs suppress APC function through a variety of means, including IL-10 expression, which depletes MHC II on their surface, TGF-β, which polarizes CD4+ T cells to Tregs, Arginase, which degrades L-argining, a valuable metabolite for T cell function, and ROS, which also impair T cell function. Neutrophils also utilize arginase and ROS to mitigate T cell function.

**Table 1 ijms-24-07359-t001:** Highlighting the effects of IR on various immune cell types. Different immune cell types from the tumor microenvironment are listed, along with examples of reported effects of IR on their function.

Cell Type Influenced	Cancer Model	Radiation Dose	Response Observed	Ref.
APC 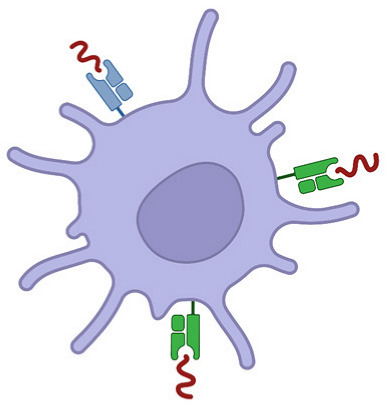	Murine pancreatic cancer tumor	4 × 6 Gy fractions	Increases in calreticulin and HMGB1 corresponded with elevated tumor-antigen presentation and increased CD8+ T cell inflitration into the tumor	[34]
	Murine colorectal tumor	20 Gy × 1 fraction	STING-dependent increases in antigen presentation by dendritic cells facilitate CD8+ T cell anti-tumor immune responses	[54]
	Murine melanoma tumor	10 Gy × 1 fraction	Depletion of dendritic cells prevented IR efficacy by preventing CD8+ T cell infiltration into the tumor	[128]
MDSC/Macrophage 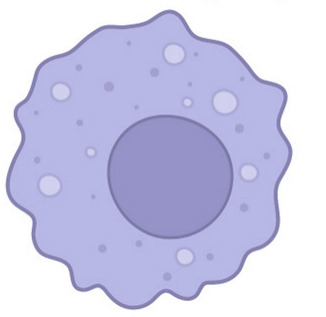	Murine colorectal tumor	20 Gy × 1 fraction	Monocytic MDSC increases in the tumor and mediates radioresistence by inhibiting CD4+ and CD8+ T cell function	[68]
	Murine prostate tumor	25 Gy × 1 fraction or 4 Gy × 15 fractions	Increases in arginase, iNOS and COX-2 expression on intratumoral macrophages	[99]
	Murine melonoma tumor	15 Gy × 1 fraction or 5 Gy × 3 fractions	Increase in macrophage infiltration to the tumor	[100]
Neutrophil 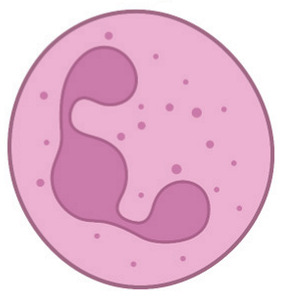	Murine soft tissue sarcoma	20 Gy × 1 fraction	Depleting neutrophils prior to IR improved treatment efficacy	[114]
	Murine prostate, lymphoma and mammary tumors	15 Gy × 1 fraction (prostate and mammary cancers) and 1.3 Gy × 1 fraction for lymphoma	Increase in neutrophil infiltration into tumors which mediate tumor cell apoptosis via ROS production	[116]
	Murine lung tumor	8 Gy × 3 fractions	Increase in neutrophil inflitration into tumors, which promotes the mesenchymal-to-epithelial transition in cancer cells via ROS production	[117]
	Murine bladder tumor	2 Gy × 1 fraction, 5 Gy × 2 fractions or 10 Gy × 1 fraction	Increased production of neutrophil extracellular traps that mediate radioresistence by inhibiting CD8+ T cell tumor infiltration	[121]
T cell 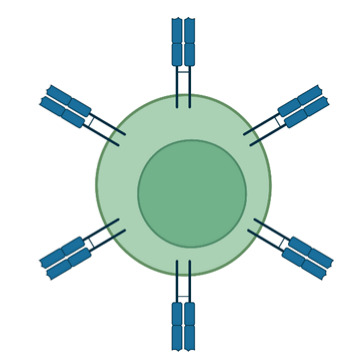	Murine melanoma tumor	10 Gy × 1 fraction	Depleting CD8+ T cells abrogates treatment efficacy	[128]
	Murine colon tumor	15 Gy in 1 fraction	Depleting CD8+ T cells abrogates treatment efficacy	[13]
	Human pancreatic cancer patients	5 Gy × 5 fractions	T cell clonal expansion in a subset of patients	[132]
	Human renal cell carcinoma patients	15 Gy × 1 fraction	T cell clonal expansion	[131]
NK cell 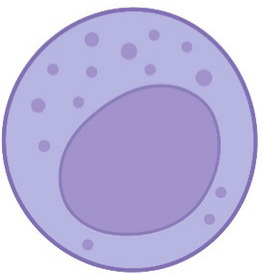	Human pancreatic cancer patients	54 Gy (median dose) × 24 fractions	Increase in NK cell infiltration into tumors	[153]
	Human melanoma cells implanted into mice	16 Gy × 1 fraction	Irradiated cancer cells prevent NK-mediated cell toxicity	[157]
B cell 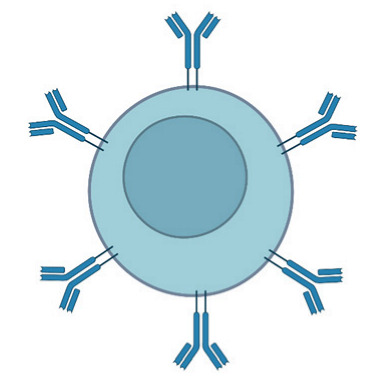	Murine colon and oral squamous cell tumors	10 Gy × 1 fraction	Increase in B cell infiltration into tumors	[172]
	Murine squamous cell tumors	12–18 Gy × 1 fraction	Increases in tumor antigen-specific B cells and depleting B cells antagonizes IR efficacy.	[173]

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
