# Peer review of "Harnessing the Immunological Effects of Radiation to Improve Immunotherapies in Cancer"

_ijms, 2023, doi:10.3390/ijms24087359_

Round 1
Reviewer 1 Report
The authors here presented a general overview of the immune response of IR. The manuscript has focused on the radiation response of different immune cells and has listed two further applications of the immune response of IR. To strengthen the link between the immune effect and IR, I suggest the authors be more specific in the following content about what has been found and discovered.
Line 83-84 ‘IR has been shown to induce these markers of ICD in a dose-dependent manner.’ The authors can specify which kind of dose range and what effect was found.
Line 127-129, ‘Regardless of the pathway 127 involved, however, multiple reports have demonstrated that Type-I IFN production is essential to IR efficacy [51, 61-63].’ Give some examples regarding IR efficacy.
A summary table of the radiation treatment regimen (dose/fraction) on the response (pro-/anti-tumor, immune stimulatory or suppressive) of different immune cells will be very helpful.
Author Response
Dear Reviewer,
Thank you for giving your time to review our manuscript.
Below are our responses to your comments. We hope we have addressed each point appropriately.
Comment 1: Line 83-84 ‘IR has been shown to induce these markers of ICD in a dose-dependent manner.’ The authors can specify which kind of dose range and what effect was found.
Response: We have now listed exactly which markers of immunogenic cell death were upregulated and specified the doses used in the papers referenced.
Comment 2: Line 127-129, ‘Regardless of the pathway involved, however, multiple reports have demonstrated that Type-I IFN production is essential to IR efficacy [51, 61-63].’ Give some examples regarding IR efficacy.
Response: We have now described that the efficacy observed was based on measures of tumor burden in these papers.
Comment 3: A summary table of the radiation treatment regimen (dose/fraction) on the response (pro-/anti-tumor, immune stimulatory or suppressive) of different immune cells will be very helpful.
Response: While it is difficult to provide a general summary on the pro-/anti-tumor effects of radiation on various cell types due to the vast differences in tumor microenvironments and radiation regimes used, we have provided a Table (Table 1) in-text that highlights immune cell-specific responses to radiation observed in the literature for various tumor models that were referenced throughout the text.
Thank you for reading,
The Authors.
Reviewer 2 Report
This manuscript deals with immunological effects of irradiation and their role in improving the antitumor immune response.
The review is well written, organized and presented. It looks at several subsets of immune cells and their involvement in reacting to the damage induced by irradiation.
I think that this works gives a concise and complete at the same time scenario of the effect of irradiation, and it is a good starting point for the reader to understand the topic.
Author Response
Dear Reviewer,
Thank you for taking the time to read through our manuscript and providing positive feedback on its content.
Kind regards,
The Authors.